# Genome-Wide Identification, Exogenous Hormone Response, Gene Structure, and Conserved Motif Analysis of the *GRF* Gene Family in *Cerasus humilis*

**DOI:** 10.3390/biology14070763

**Published:** 2025-06-25

**Authors:** Lingyang Kong, Lengleng Ma, Shan Jiang, Xinyi Zhang, Junbai Ma, Meitong Pan, Wei Wu, Weili Liu, Weichao Ren, Wei Ma

**Affiliations:** Pharmacy of College, Heilongjiang University of Chinese Medicine, Heping Road, Harbin 150040, China; hljkly970219@163.com (L.K.); 18739466784@163.com (L.M.); 17390928032@163.com (S.J.); zhangxinyi121102@163.com (X.Z.); 15114516116@163.com (J.M.); meitong_pan@163.com (M.P.); wuwei52414@163.com (W.W.); liuweili410@126.com (W.L.)

**Keywords:** *Cerasus humilis*, *GRF* gene, gibberellin treatment, qRT-PCR, expression patterns

## Abstract

GRF, a transcription factor unique to plants, is crucial for plant growth and development as well as for reacting to environmental pressures. This encompasses treatments for cold, oxidation, osmotic stress, and hormones to control growth and development. The medicinal benefits of *C. humilis* are immense, earning it widespread affection and trust. Despite reports of the *GRF* gene in numerous species, research on *C. humilis* remains inadequate. The research not only verified the comprehensive bioinformatics examination of every *GRF* gene throughout *C. humilis*’s genome. Concurrently, the expression levels of the *ChGRF* gene in different tissues were studied, and the expression of the *ChGRF* gene under gibberellin treatment was also explored. These results provide a theoretical basis for understanding the function of *GRF* genes.

## 1. Introduction

Transcription factors play an essential role in the growth, development, metabolism, reproduction, and differentiation of plants [1]. Growth-regulating factors (*GRF*) are a family of small transcription factors that play an important role in plant growth, development, and lifespan [2,3,4,5]. *GRF* genes are an important class of genes. They have two conserved domains at the N-terminus, namely, QLQ (Gln, Leu, Gln) and WRC (Trp, Arg, Cys) at the N-terminus, respectively [6]. Resembling the N-terminal end of SWI2/SNF2 in yeast, the SWI2/SNF2 protein structure participates in chromatin remodeling complexes in yeast, suggesting that the QLQ domain may also have this function [7]. Furthermore, the QLQ domain can engage with the preserved configuration of the GIF protein SNH to execute the transcription activation process. In the WRC domain, along with the zinc finger motif involved in DNA binding, there is a nuclear localization signal (NLS). Binding to the cis-acting regions of target genes, it can regulate the expression of downstream genes [8]. *OsGRF1* is the first *GRF* gene cloned from *Oryza sativa* L, and it plays an important role in the regulation of stem growth [9]. On this basis, *GRF* transcription factors in a variety of plants have been identified and analyzed.

The *GRF* gene family is involved in various physiological activities, especially at the early stage. These include leaf development [10], root growth [11], floral organ formation [12], seed formation [13], hormonal signaling, etc. [14]. Among them, plant hormones include auxin, cytokinin (CK), abscisic acid (ABA), gibberellin (GA), etc., and they play roles by responding to the signal transduction of hormones [15]. Strong expression of the *GRF* family is evident in the tips of roots, buds of flowers, and immature leaves. Elevating the levels of *BnGRF2a* can boost Brassica napus oil production by 50%, accelerate leaf growth by 20%, and enhance photosynthesis efficiency by 40%. The overexpression or disruption of *AtGRF5* will affect the size of *A. thaliana* leaves. The function of *AtGRF5* is more crucial in the processes of growth and development [16]. *AtGRF8* participates in the development of plant flowers. Furthermore, in *A. thaliana,* the overexpression of the *BnGRF2a* gene promotes the elongation of the pistil, resulting in a low selfing rate. Therefore, deep research on the *GRF* gene family of *C. humilis* is fundamental.

*C. humilis*, also known as Chinese dwarf cherry, is a dwarf fruit tree belonging to the genus Cerasus in the Rosaceae family. The appearance of its fruits is identical to that of cherries, and the taste is similar to that of plums. They are sweet and sour, quite delicious, and possess a unique flavor and fruit fragrance [17,18]. The pulp of *C. humilis* is rich in nutrients such as sugar, protein, mineral elements, vitamins, and amino acids. In particular, it has high contents of calcium and iron. The calcium and iron content in every 100 g of fresh fruits can reach 60 mg and 1.5 mg, respectively, seven and six times that of apples. Moreover, they are mainly water-soluble calcium and calcium phosphate, and therefore easily absorbed by the human body [19]. These nutritional components make *C. humilis* an ideal choice for health-conscious people. Nonetheless, the conveyance and preservation of *C. humilis* may be influenced by non-living environmental factors like temperature. Therefore, choosing appropriate methods to improve the quality of fruits becomes exceptionally crucial. Gibberellin (GA), as a hormone originating from within plants, is generally regarded as a key signaling molecule in plants. GA, a plant growth regulator, is extensively utilized in agriculture, significantly aiding in seed sprouting, stem extension, fruit growth, and enhancing the plant’s resilience to stress. Therefore, in this study, *C. humilis* was treated with 0.5 mmol L^−1^ GA, and the deterioration of the fruits under different treatment times was observed, aiming to provide new ideas for the storage and transportation of *C. humilis*. There is increasing evidence that the *GRF* family is also involved in abiotic stress response [20,21]. However, no research has been reported on the connection between the *GRF* gene of *C. humilis* and GA treatment. Therefore, we identified the *ChGRF* gene family of C. humilis and further investigated the changes in the expression of the *C. humilis GRF* genes under low-temperature GA treatment.

To this end, the project selected *C. humilis* as material and conducted functional studies of the *GRF* genes. It is speculated that gibberellin has a delaying effect on the excessive ripening of *C. humilis* and provides a preservation effect. With the rapid development of genome sequencing technology, the *GRF* family has been reported in recent years. Based on earlier work, this project aims to investigate the response of *ChGRFs* to exogenous hormones (gibberellin) via whole-genome sequencing. This project plans to further clarify the *GRF* gene family in *C. humilis* by researching aspects such as the structure, evolutionary pattern, chromosomal location, collinearity relationship, and expression pattern of the *GRF* genes in *C. humilis*. These results may be helpful for further research on *GRF* function in *C. humilis*.

## 2. Materials and Methods

### 2.1. Plant Materials

Genome sequencing of *C. humilis* was downloaded from the National Genomics Data Center [22], accessed on 25 December 2024. The GRF protein sequence was downloaded from the TAIR database (https://www.arabidopsis.org/, accessed on 28 December 2024). Access to the GRF protein sequences of *Arabidopsis thaliana* was obtained from The Arabidopsis Information Resource (https://www.arabidopsis.org (accessed on 28 December 2024)), rice sequences from the Rice Genome Annotation Project (http://rice.plantbiology.msu.edu/analyses_search_locus.shtml (accessed on 28 December 2024)), Chinese cabbage sequences from the Brassica database (http://brassicadb.org/brad/ (accessed on 28 December 2024)), rapeseed sequences from BrassicaDB (http://www.brassicadb.cn (accessed on 28 December 2024)), poplar and maize sequences from Phytozome (https://phytozome.jgi.doe.gov/pz/portal.html (accessed on 28 December 2024)), and strawberry sequences from the Genome for Rosaceae (https://www.rosaceae.org (accessed on 28 December 2024)). In this study, the samples were taken from the healthy *C. humilis* in the orchard of Heilongjiang Province, China. This study used healthy dwarf cherries from the Medicinal Plant Garden of Heilongjiang University of Traditional Chinese Medicine as experimental materials. The fruits were immersed in a 0.5 mmol L^−1^ GA [23,24] solution, stored at 4 °C, and collected after treatment for 6 h, 12 h, 24 h, 3 d, and 5 d, and repeated the biological process three times to confirm the reliability of the experimental data. The samples were immersed in liquid nitrogen and kept at −80 °C, awaiting RNA extraction. (GA purchased from Beijing Kulaibo Technology Co., Ltd. in China (item number: CG5571)).

### 2.2. Identification and Characterization of ChGRF Genes

We extracted the CDS [25] sequences of these data using the following formula TBtools-II (v2.136) and convert it into protein sequences, submitted to Plant Transcription Factor Database (PlantTFDB, http://planttfdb.gao-lab.org/index.php, accessed on 28 December 2024) for transcription factor prediction. During candidate sequencing, to further screen for conservative structural domains, we used the CD search tool (https://www.ncbi.nlm.nih.gov/Structure/cdd/wrpsb.cgi, accessed on 28 December 2024) and Pfam database (https://ngdc.cncb.ac.cn/, accessed on 28 December 2024). Download hidden Markov files of WRC (PF08879) and QLQ (PF08880) domains to validate the *ChGRF* gene family, manually remove redundant sequences, retain the longest gene sequence, and ultimately identify the sequence as a candidate member of the *ChGRF* gene family. Predicting the physicochemical properties of proteins using the ExPASy website (http://web.expasy.org/protparam/, accessed on 28 December 2024) [26], including the isoelectric point (PI), molecular weight (MW), and chromosomal position (CP) of the proteins of *ChGRFs*.

### 2.3. Phylogenetic Analysis of the ChGRF

In order to gain a better understanding of the *ChGRF* gene family, the full-length amino acid sequences of nine AtGRF proteins from *A. thaliana* and twelve ChGRF proteins from *C. humilis* were aligned using ClustalX (2.1). Based on the following parameters, the MEGA-X11.0.11 [27] software was used to construct the phylogenetic tree of the *ChGRF* gene family system.

### 2.4. Collinearity and Evolutionary Gene Analysis

Necessary data for pinpointing the *ChGRF* gene family’s chromosomal location, encompassing initial and final positions, chromosome length, and count, were acquired through the TBtools-II (v2.136) software in *C. humilis* genome annotation file. The Circus of TBtools-II was utilized to draw the gene chromosomes. MCScanX (V1.0.0) [28] software was used to identify the repetitive sequences of the *GRF* gene and conduct visual analysis using TBtools-II.

### 2.5. Gene Structure and Motifs of ChGRF

The *ChGRF* sequences were submitted to MEME (https://meme-suite.org/meme/tools/meme, accessed on 28 December 2024) [29]. The *ChGRF* motif was predicted, and the position distribution of each sequence was selected. The number of motifs preserved was fixed at 10, while all other variables were adjusted to their standard values. The TBtools-II software’s Gene Structure View feature was employed, merged with the phylogenetic tree, to display *C. humilis’* annotation files. The files encompass coding and non-coding segments of the gene architecture, along with the internal exons and preserved domains.

### 2.6. Cis-Element Analysis and GO Analysis of the ChGRF Promoter

In this study, the *ChGRF* gene initiation codon (ATG) was extracted and presented to PlantCare (http://bioinformatics.psb.ugent.be/webtools/plantcare/html/, accessed on 28 December 2024). On the website, cis-acting elements were forecasted using standard settings, and the outcomes of Plantcare analysis were streamlined by incorporating categories related to stress, hormones, and growth. Visualization was conducted using TBtools-II software, while the Evolview website served the purpose of beautification. GO analysis of ChGRF protein sequences (http://www.bioinformatics.com.cn/, accessed on 28 December 2024). The results are displayed on the website.

### 2.7. Protein–Protein Interaction Network Analysis

The prediction of interactions between *C. humilis* GRF proteins used the online STRING website (https://cn.string-db.org/, accessed on 28 December 2024). The information is stored in a TSV format and is rendered visually using the Cytoscape software (3.7.2) (www.cytoscape.org). Moreover, we analyzed the three-dimensional structure of the *C. humilis GRF* genes and visualized them using the Swiss model website (SWISS-MODEL Interactive Workspace) (https://swissmodel.expasy.org/interactive/, accessed on 28 December 2024).

### 2.8. Expression Analysis of ChGRF Genes in Different Tissues and Developmental Stages

To investigate the differences in the expression patterns of *GRF* in various tissues, the previously published transcriptome data of *C. humilis* were used, including the expression in different tissues and at different fruit developmental stages. The transcriptome data were transformed using the Log2 (FPKM + 1) algorithm to determine the gene expression levels. Then, the TBtools-II software was applied to visualize the results of *GRF* gene expression.

### 2.9. RNA Extraction Andq qRT-PCR Analysis

Take out the fruits of *C. humilis* frozen in the −80 °C refrigerator and extract RNA at time intervals after storage. According to the instructions, use the plant total RNA extraction kit (Simgen Biotechnology Co., Ltd., Hangzhou, China). Then, the Surescript TM First Strand cDNA Synthesis Kit (Simgen Biotechnology Co., Ltd., Hangzhou, China) is used to reverse transcribe the total RNA. The test is then performed with a QPCR kit, BlazeTaqTM SYBR ^®^ Green QPCR-Mix 2.0 (Guangzhou, China), and the primers are listed in Appendix A. The PCR procedure comprises heating for 30 s at 95 °C, then heating for 45 s at 95 °C for 12 s, heating at 58 °C for 30 s, heating at 58 °C for 45 s, and finally for one second at 79 °C. Following a last PCR cycle, raise the temperature at 0.5 °C to 99 °C from 55 °C to 99 °C to create a melting curve for the sample [30]. Internal reference proteins are used as reference genes, and the 2^−∆∆Ct^ technique is employed to measure and compare gene expression [31]. Each sample was tested three times.

## 3. Results

### 3.1. Identification of GRF Genes

To determine the *GRF* gene sequences of *C. humilis*, the Hidden Markov Model search (HMMER search) method was adopted. The Hidden Markov Models PF08880 (QLQ domain) and PF08879 (WRC domain) were used to verify the candidate *GRF* genes of *C. humilis*. BLASTP searches were conducted using the GRF proteins of *A. thaliana*. As a result, twelve *ChGRF* genes were identified, and each gene contained the GRF domain. The names of *ChGRF* were designated as *ChGRF1*–*ChGRF12*, originating from their distribution on seven chromosomes. Furthermore, we have extended our analysis to the physical and chemical properties of twelve *ChGRF* proteins. The sequence length ranged from 331 to 1034, and the molecular weight ranged from 36,302.21~117,919.96. The isoelectric point ranged from 5.56 to 9.29. The basic information of each member of the *ChGRF* family is concisely presented in Appendix A.

### 3.2. Phylogenetic Analysis

For a more thorough confirmation of the evolutionary connections among *GRF* genes, a phylogenetic tree was constructed by comparing the full-length amino acid sequence alignments of nine identified *A. thaliana GRFs*, twelve rice (*Oryza sativa*) *GRFs*, several *Fragaria vesca GRFs*, nineteen *poplar GRFs*, fifteen *Brassica napus GRFs*, seventeen *Brassica rapa GRFs*, and thirty-five rapeseed *GRFs* (Appendix A). The phylogenetic tree is divided into five subgroups (A–E) based on phylogenetic and species evolutionary relationships. Among them, the GRF proteins of *C. humilis* were split into five lineages. *ChGRF1* and *ChGRF2* belonged to the D family, *ChGRF5* and *ChGRF11* belonged to the C family, *ChGRF7* belonged to the B family, *ChGRF3*, *ChGRF4,* and *ChGRF9* belonged to the E family, and *ChGRF6*, *ChGRF8*, *ChGRF10*, and *ChGRF12* belonged to the A family (Figure 1).

### 3.3. Chromosome Localization Analysis

To further study the distribution of the *ChGRF* gene, this study located all members of the *ChGRF* gene family on the chromosomes (Figure 2). The *ChGRF* genes are unevenly distributed on each chromosome. Twelve *ChGRF* genes are evenly distributed on seven chromosomes. Among them, Chr3 contains the most significant number of *ChGRF* genes (three members), followed by Chr2, Chr4, and Chr7, each with two *ChGRF* members, and finally, Chr1, Chr5, and Chr6 each contain one member of the *ChGRF* gene family. In addition, according to our research findings, there is no direct connection between the length of chromosomes and the number of *GRF* genes. This distribution pattern of chromosomes is similar to that of the *GRF* genes in strawberry [32].

### 3.4. Collinearity and Evolutionary Analysis

As can be seen from the figure, in the genome of *C. humilis*, six *ChGRF* genes are located on the duplicated segments, and they are paired with three pairs of segmental duplications (*ChGRF1* and *ChGRF2*), (*ChGRF5* and *ChGRF11*), (*ChGRF8* and *ChGRF10*) (Figure 3A). These duplicated segments are the main driving forces for the evolution of *ChGRF* genes [33]. In addition, the *ChGRF* duplicated genes are evenly distributed among subfamilies A, C, and D, indicating that these chromosomal segments may not have undergone complete differentiation during the evolutionary process and may be functionally redundant or superfluous [34].

To conduct a more in-depth study on the occurrence and development of genes in the *ChGRF* family, we compared three dicotyledonous plants, namely, *A. thaliana*, *Malus pumila* Mill, and grapevine, with one monocotyledonous plant, *Oryza sativa* L. There are six collinear genes between *ChGRF* and *AtGRF*, twelve pairs between *ChGRF* and *MmGRF*, eight pairs between *ChGRF* and *VvGRF*, and three pairs between *ChGRF* and *OsGRF* (Figure 3B). Compared with monocotyledonous plants, the number of collinear genes in dicotyledonous plants was higher than that in monocotyledonous plants, among which *Malus pumila* Mill had a relatively large number of collinear genes.

### 3.5. Analysis of the GRF Gene Structure and Conserved Motifs

To better understand the genetic structure and distribution of *GRF* genes in *C. humilis*, we conducted a detailed study on the gene structures of all twelve *GRF* genes by using the GSDS online server (Figure 4A). We investigated the intron–exon composition of the 12 *ChGRF* genes (Figure 4B). The count of introns in the *ChGRF* genes varied between 0 and 14. Among them, the *ChGRF3* gene contained 14 introns, while most genes contained 3 introns, indicating that some introns in the *ChGRF* genes might have been lost during the evolutionary process. For example, *ChGRF10* in subfamily A and *ChGRF4* in subfamily E did not contain untranslated regions (UTRs), while *ChGRF2*, *ChGRF5*, and *ChGRF11* had two coding sequence (CDS) regions and two UTR regions. It was speculated that some members of this subclass might have undergone gene splicing or gene fragment insertion in the evolutionary process [25].

Meanwhile, we also studied their conserved structures. We analyzed 10 representative motifs using the MEME software (5.5.8), selected 10 conserved motifs (Motifs 1–10) from them, and located them in each gene. Our previous studies found that, except for *ChGRF9*, all the other *GRF* genes contain Motif 1 and Motif 2. The *ChGRF1* gene contains Motif 3, while the *ChGRF3* gene contains Motif 6. These three segments form a classic *GRF* gene family. We speculate that most members in the same branch have similar motifs, indicating that homologous proteins may play similar roles during the evolutionary process. The results of this study further prove that the *ChGRF* gene family can be divided into different clusters.

### 3.6. Cis-Elements of ChGRF Genes and Their Functional Analysis

Forecasting of cis-acting elements in the 2000 bp segment before the *GRF* gene family in *C. humilis* revealed the presence of several TATA boxes and CAAT boxes in *ChGRF*, suggesting standard transcription. Using the PlantCRAE website, we successfully identified the cis-elements within the *ChGRF* genes. We then synthesized and outlined their potential functions, visualizing this information through a comprehensive plot. (Figure 5A). Furthermore, the promoters of the *ChGRF* genes contain numerous cis-regulatory components. Our analysis pinpointed 302 cis-acting elements, which can be categorized into four groups: those associated with development, environmental stress-related elements, and light-responsive elements (Figure 5B,C). The B-binding site associated with photoreactivity is a major hormone-responsive element that can react with abscisic and salicylic acid. In the upstream *ChGRF* gene family, the ABA is the most abundant in hormone-related cis-elements. The next largest group is light-sensitive elements with 121 elements (40.1%). The third category consists of elements associated with environmental stress, comprising 63 elements (20.8%). The fourth group is development-related, with 29 elements (9.6%).

### 3.7. GO Analysis of the ChGRF Gene Family

GO conducted functional verification on twelve *ChGRF* genes. Previous studies have found that the *ChGRF* genes possess ten major functional components, molecular functions, and biological processes. During this process, hormones, light, and low temperature can all regulate them. The *ChGRF* genes are mainly located in the Golgi apparatus cytoplasm, that is, within the chloroplasts. In terms of molecular functions, these genes can bind to proteins both at the post-transcriptional and molecular levels and can also assist in their expression (Figure 6). 

### 3.8. Protein–Protein Interaction Network Analysis and Homology Modeling

Using the STRING database, we predicted the protein–protein interaction characteristics of the *ChGRF* family. In Figure 7A, a node represents the name of a protein. When the number of nodes connected to a node increases, its degree value will become larger. The node’s dimensions and the color’s depth signify the degree value. Studies have shown that *ChGRF1* is highly expressed and interacts strongly with other proteins. These results suggest that this protein is vital in regulating plant growth.

The SWISS-MODEL interactive workspace website was used to predict the twelve types of *ChGRF* further. The results showed that the RMSD for the majority of ChGRF proteins after superposition was below 1, suggesting the reliability of the prediction (Figure 7B). Therefore, the study of the 3D spatial structure of *ChGRF* is critical to elucidate its biological function.

### 3.9. Analysis of ChGRF Gene Expression

To delve deeper into the possible functions of *ChGRF* genes, our research focused on examining their expression across other tissues and growth phases. Figure 8A shows the expression profiles of the *ChGRF* genes in different tissues. Among them, six members (*ChGRF1*, *ChGRF2*, *ChGRF3*, *ChGRF8*, *ChGRF9*, *ChGRF12*) had high expression levels in flowers. Eight members had high expression levels in fruits (*ChGRF1*, *ChGRF2, ChGRF3*, *ChGRF8*, *ChGRF9*, *ChGRF10*, *ChGRF11*, *ChGRF12*). Two genes are highly expressed in leaves (*ChGRF3*, *ChGRF9*). *ChGRF3*, *ChGRF9*, *ChGRF10*, and *ChGRF12* had high expression levels in roots. *ChGRF1*, *ChGRF2*, *ChGRF3*, *ChGRF9*, *ChGRF10*, and *ChGRF12* had high expression levels in stems. *ChGRF4*, *ChGRF5,* and *ChGRF6* were expressed in all tissues; however, their expression levels were relatively low. While *ChGRF7* was not expressed in the tissue parts shown in the figure below, this does not mean that it is not expressed. This could manifest in different tissues or throughout various growth and development phases. The expression patterns of *ChGRF* genes during various developmental phases of *C. humilis* are depicted in Figure 8B. There were three replicates at each developmental stage. A total of eight members (*ChGRF2*, *ChGRF4*, *ChGRF5*, *ChGRF6*, *ChGRF8*, *ChGRF10*, *ChGRF11*, *ChGRF12*) had higher expression levels in the young fruit stage than in the other three stages. The findings suggest that these genes are crucial in the growth and maturation of *C. humilis* fruits, having developed distinct regulatory mechanisms.

### 3.10. RNA Extraction and qRT-PCR Analysis

To investigate the role of *ChGRF* genes, the effect of exogenous hormone GA on the expression of *ChGRF* was investigated, resulting in twelve specific *ChGRF* genes being analyzed. A comparative analysis was conducted on *C. humilis* fruits that underwent GA treatment for 0 h, 6 h, 12 h, 24 h, 3 d, and 5 d. The results showed that as the storage time prolonged, the degree of surface folding of *C. humilis* fruits increased and the degree of fruit softening became more severe (Figure 9).

Based on this heat map, screening and analysis were carried out using qRT-PCR (Figure 10). The impact of gibberellin on the preservation effect of apple fruits was studied. The expression of *ChGRF11* and *ChGRF12* was first reduced, then increased, while *ChGRF1*, *ChGRF2*, *ChGRF6*, and *ChGRF8* tended to rise and decrease. *ChGRF9* and *ChGRF10* were highest at 0 h. Through verification using qRT-PCR, we found that the *ChGRF* genes showed differential expression during the gibberellin response process and might cooperate in development (Appendix A).

## 4. Discussion

Not only can the fruit of *C. humilis* be eaten fresh, but it can also be made into wine, fruit juice, jam, and sweets [35]. In addition, due to its high content of flavonoids, the fruits of *C. humilis* can serve as an effective natural antioxidant and prevent damage to the human body caused by free radicals to varying degrees. It has high application potential in the medical and healthcare industries [36]. Therefore, studying and analyzing the transcription factors related to fruit quality is essential. The plant-specific *GRFs* act as attractants and play a key role in regulating the different stages of plant growth and development and in a number of aspects of the response to stress. There has been no research on the *GRF* genes of *C. humilis*. A comprehensive analysis of *ChGRFs* was conducted, focusing on their expression changes in fruits during the storage process and after treatment with the GA hormone.

GA3 enhances seed sprouting and affects protein metabolism through the regulation of hydrolysis in embryonic reserve tissues. When present in sufficient amounts in seeds, GA3 enhances cell growth by prompting the primary roots to fracture tissues that hinder their development [37]. Similarly, gibberellins affect protein metabolism by creating enzymes that compromise the seed coat, playing a part in the growth and extension of the primary root.

Santos and colleagues (2016) noted enhanced vitality in Passiflora alata Curtis seeds when submerged in GA3 solution at 500 mg/L and 1000 mg/L levels. Additionally, gibberellins neutralize the suppression caused by abscisic acid, leading to a natural rise in GA3 levels, showcasing their function in breaking down seed dormancy and improving the physical condition of seedlings [37]. This has been confirmed in grape varieties such as Thompson Seedless [24] and Syrah [38]. Moreover, the application of red elements can improve the quality of pre-grape fruits, enhance their commercial value, and increase economic benefits.

Twelve *ChGRFs* in *C. humilis* were identified. We constructed the phylogenetic trees of ChGRF, AtGRF, BnGRF, BrGRF, PtGRF, FvGRF, ZmGRF, and OsGRF proteins. Based on the 12 kinds of ChGRF proteins and 9 kinds of AtGRF proteins, they were divided into five groups (Groups A–E). The ChGRF proteins were evenly distributed among these five groups. From a phylogenetic point of view, an analysis of the subfamily standings of various *ChGRF* gene family members revealed significant homology and structural resemblance among those grouped in the same branch [2]. It has been shown that *ChGRF1* and *ChGRF2* are homologous to *AtGRF1* and are highly expressed in the roots. The expression of *OsGRF10* was high in the rice leaf [39]. *ChGRF12*, *ChGRF10,* and *ChGRF6* were found to be homozygous and highly expressed in leaves. It is speculated that this group has relatively high functions in leaves. Under normal circumstances, the expression level of *GRF* genes in vigorously growing tissues is higher than that in mature tissues [7]. This indicates that, as both are dicotyledonous plants, their taxonomical conclusions are basically the same. This suggests that *ChGRF* has good gene structure stability. Despite some differences, the phylogenetic tree is mainly consistent with the results of earlier studies [40]. Genes with similar functions tend to be classified into the same subclass, which offers a crucial basis for predicting gene functions [25].

Gene duplication promotes the generation and function of new genes, significantly influencing the evolutionary process. During the evolutionary process, there are three ways of gene duplication: segmental duplication, tandem duplication, and translocation events. In expanding plant gene families, segmental and tandem duplication are mainly observed. Chromosome localization analysis revealed their irregular distribution on seven chromosomes. A total of six segmental duplications were identified, while there were only three tandem duplications between *ChGRF1* and *ChGRF7*, *ChGRF5* and *ChGRF11*, and *ChGRF8* and *ChGRF10* [20,41]. Changes in the upstream and the coding regions may influence the function of the duplicated genes [42]. Comparative studies of the *C. humilis* genome with the genomes of *Arabidopsis thaliana*, apple, grape, and rice plants have shown significant collinearity between *C. humilis GRF* and dicotyledonous plants.

The configuration of genes and the spread of motifs offer corroborative proof for the evolutionary links between various species or genes. The analysis of amino acid sequences indicated that the *ChGRF* genes shared the same gene structure and motif, suggesting that the *GRF* had similar functions in the same branch [43,44]. Besides the *GRF* domains, ten different motifs were also found, which are distributed differently in *ChGRF*. There is a close relationship between intron differentiation and the evolution of plants. There are often similar exon–introns and quantificational distributions in the same subfamily.

In the promoter region, cis-acting elements are essential to control gene expression [45] and play a key role in controlling the expression of downstream associated genes during the transcription phase, thereby increasing the plant’s resistance to pressures from the outside world. At 2000 bp upstream of the promoter, we identified a number of *ChGRF* responses.

*GRF* is key in synthesizing plant hormones, stress response, and regulation. Adverse environmental stress is a significant constraint on crop growth and the improvement of agricultural product quality [46]. As a key regulatory protein, *GRF* plays an essential role in regulating hormonal signals [47]. The growth regulator (*GRF*) in tobacco is a key transcription factor in GA biosynthesis [15]. We analyzed the *ChGRF* sequences and found multiple hormone-regulatory elements in *ChGRFs*, suggesting that *ChGRFs* may play an essential role in hormone signal transduction [48].

More and more evidence has emerged that genes play an essential role in plant development [49]. Generally, gene expression patterns are key indicators for predicting gene functions [50]. The GA signaling pathway plays an important regulatory role in plant stress conditions. Among the *GFR* genes of *C. humilis*, *ChGRF2* reached the highest expression level 3 d after GA treatment, *ChGRF3* reached the highest level at 24 h, *ChGRF6* had the highest expression at 3 d, and *ChGRF4*, *ChGRF11,* and *ChGRF12* reached the highest levels at 5 d. Based on previous speculation, we can conclude that gibberellin specifically delays excessive fruit ripening and preservation. Through fluorescence quantitative PCR, it was found that the delay in excessive maturation under gibberellin treatment may be due to the role of *ChGRF* genes. Therefore, our study is very necessary to provide a theoretical basis for understanding the function of *GRF* genes.

## 5. Conclusions

This research pinpointed 12 individuals belonging to the *ChGRF* gene family for the first time. These elements are spread across seven chromosomes in the *C. humilis* genome, each housing QLQ and WRC domains. Based on the evolutionary connections among species, it was segmented into five distinct categories. Family members exhibit preserved exon–intron configurations, motifs, and categories. At the same time, gibberellin has a preservative effect on *C. humilis* fruit, and it is further speculated that *ChGRF2*, *ChGRF6*, and *ChGRF12* may play a key role. In summary, this is a preliminary study of the *GRF* gene family in *C. humilis* fruits, providing a reference for further investigating the functions of *GRF* genes in different species and expanding our understanding of the molecular mechanisms of *C. humilis* fruit’s response to gibberellin.

## Figures and Tables

**Figure 1 biology-14-00763-f001:**
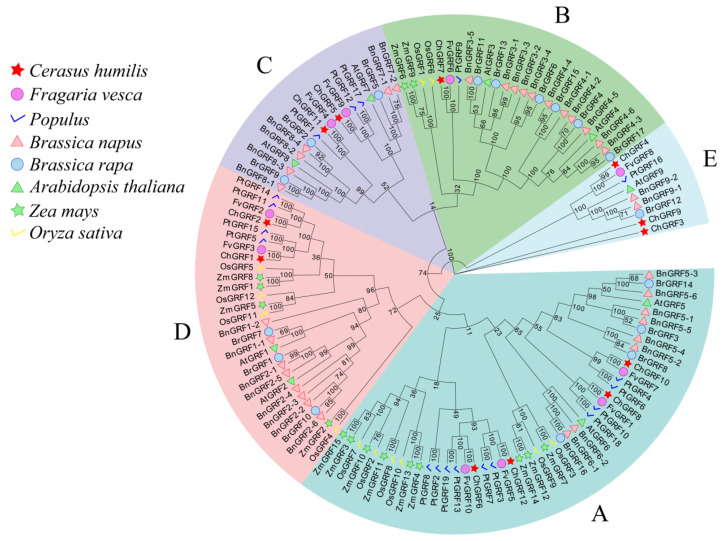
Phylogenetic and classification analysis of GRF proteins in *C. humilis*, *Fragaria vesca*, poplar, *Brassica napus*, *Brassica rapa*, *A. thaliana*, *Zea mays,* and *Oryza sativa*. A. B, C, D, and E represent five different subfamilies. Branches are represented by different colors to indicate *GRF* subfamilies. *C. humilis* is indicated by a red five-pointed star; *strawberry* is represented by a pink circle; *poplar* is indicated by a blue tick; rapeseed is represented by a light-colored triangle; *Chinese cabbage* is represented by a blue circle; *A. thaliana* is represented by a green triangle; maize is represented by a green triangle; and *rice* is represented by a yellow tick.

**Figure 2 biology-14-00763-f002:**
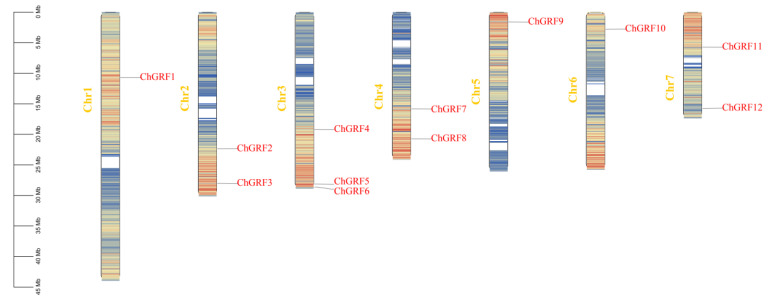
Distribution of the *ChGRF* gene in *C. humilis* genome. The chromosome localization of each *ChGRF* gene was determined according to the genome assembly of *C. humilis*. Each chromosome’s number is displayed atop the chromosome.

**Figure 3 biology-14-00763-f003:**
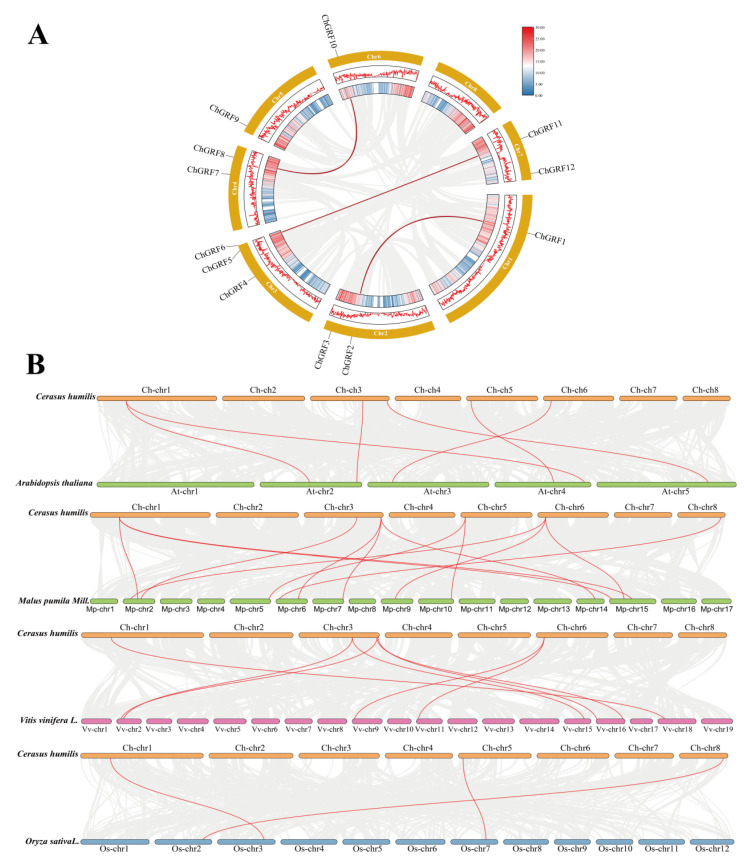
Chromosome localization and interspecific collinearity analysis: (**A**) The grey lines represent all collinearity genes in the *C. humilis* genome, and the red line represents the tandem replication line relationship between *ChGRF* genes. (**B**) Collinear relationship between *C. humilis* and other species. The red lines represent collinear gene pairs.

**Figure 4 biology-14-00763-f004:**
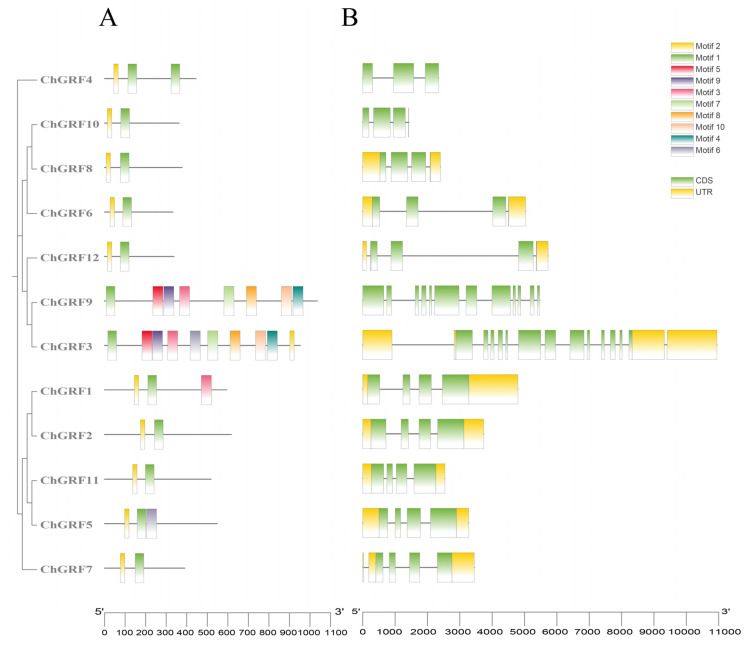
Phylogenetic and conserved motif analysis of *ChGRF*: ((**A**) *ChGRF* conserved motif analysis; (**B**) exon–intron structure analysis. Yellow blocks, black lines, and green blocks, respectively, denote exons, introns, and non-coding region).

**Figure 5 biology-14-00763-f005:**
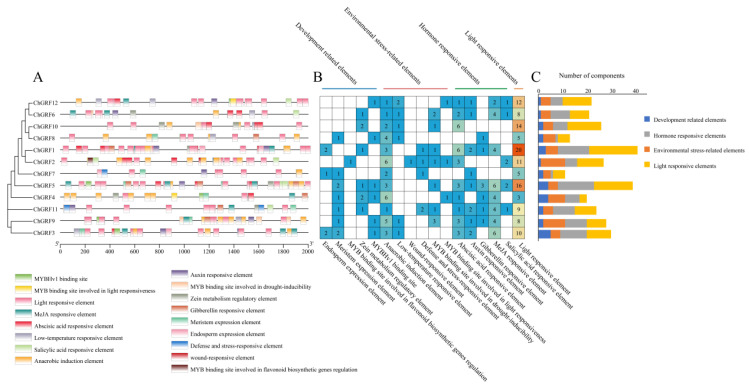
Analysis of cis-regulatory elements of *ChGRF* genes in *C. humilis*: ((**A**) Distribution of cis-acting elements in the promoter region of the *ChGRF* gene. (**B**,**C**) *ChGRF* gene promoter. Yellow represents light response elements, orange represents environmental stress-related elements, gray represents hormone response elements, and blue represents development-related elements).

**Figure 6 biology-14-00763-f006:**
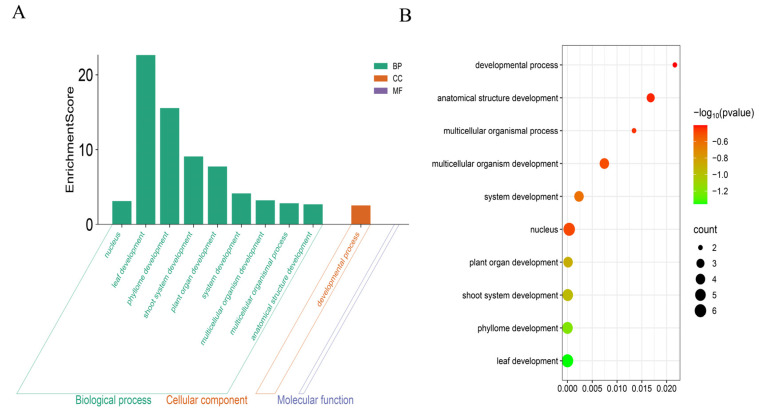
GO annotations of members of the *ChGRF* gene family: (**A**) green represents biological processes, orange represents cellular components, and purple represents molecular functions. (**B**) GO term enrichment analysis of *ChGRF* genes.

**Figure 7 biology-14-00763-f007:**
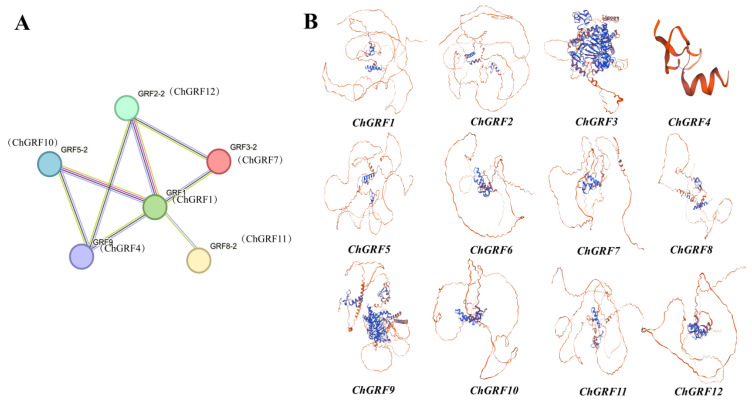
Protein interaction analysis and three-dimensional structure diagrams of *ChGRF* genes: (**A**) ChGRF protein interaction network; (**B**) three-dimensional structure of *ChGRF* gene.

**Figure 8 biology-14-00763-f008:**
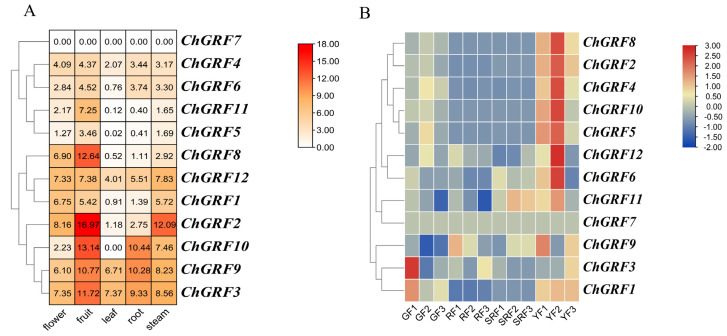
Analysis of *ChGRF* gene expression: (**A**) The expression of *ChGRF* in different tissue sites. (**B**) The different manifestations of *ChGRF* in fruits of different colors (GF: green fruit; RF: red fruit; SRF: small red fruit; YF: yellow fruit).

**Figure 9 biology-14-00763-f009:**
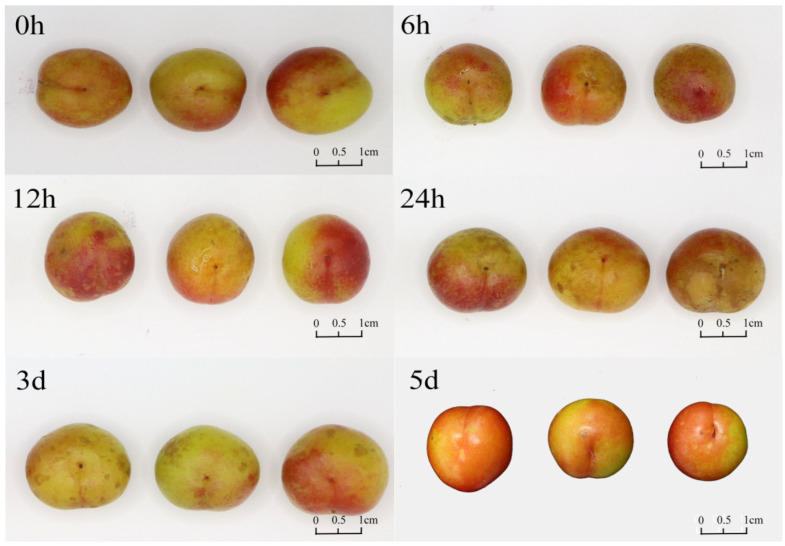
Phenotypic characteristics analysis of *C. humilis* under gibberellin treatment (0 h, 6 h, 12 h, 24 h, 3 d, 5 d).

**Figure 10 biology-14-00763-f010:**
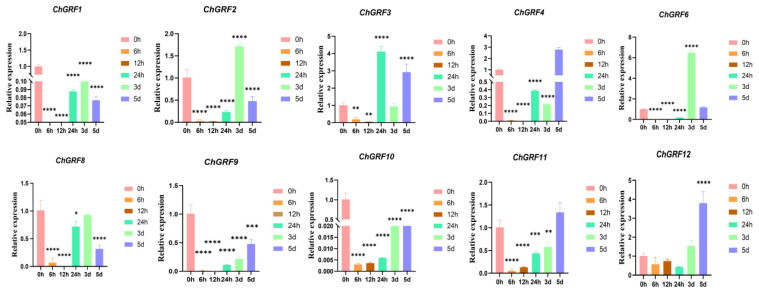
qRT-PCR analysis of *ChGRF* genes under GA hormone treatment. One-way analysis of variance (ANOVA) was used to analyze the significance of copper stress treatment for 0 h, 6 h, 12 h, 24 h, 3 d, 5 d, and CK samples (* *p* < 0.05; ** *p* < 0.01, *** *p* < 0.001, **** *p* < 0.0001).

## Data Availability

All data generated or analyzed in this study are included in the main text and its Appendix A.

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
