# Peer review of "Genome-Wide Identification, Exogenous Hormone Response, Gene Structure, and Conserved Motif Analysis of the GRF Gene Family in Cerasus humilis"

_biology, 2025, doi:10.3390/biology14070763_

Round 1
Reviewer 1 Report
Comments and Suggestions for Authors
Dear,
I have reviewed the manuscript, which presents a comprehensive study on the GRF gene family in response to exogenous hormones in Cerasus humilis. The methodology appears sound, and the results are well presented. Below is a list of reviewed comments.
Language and Grammar/Phrasing: The English is generally understandable, but there are numerous instances of grammatical errors, awkward phrasing, and typos that hinder readability. Professional English editing would significantly improve the manuscript.
Figures: Please add the detailed captions in the title of each figure for clear readability.
Data-retrieving information: mention the exact date of retrieving the data from different databases.
Write the complete names and versions of all software used for analysis.
Write the exact genome versions of all the crops used for evaluating the phylogenetic association.
Scientific names and gene names should be italicized.
Abstract: Please mention the names and IDs of candidate genes and encoding proteins.
In the Plant Material section, please add the complete details of exogenous GA hormone treatments.
Discussion: It needs to be expanded by adding the more informative and comparative literature about genes' regulatory function in response to the GA treatments in other crops.
Comments on the Quality of English LanguageThe English could be improved by extensive editing to more clearly express the research.
Author Response
Dear Editorand Reviewers,
We respectfully submit our original research article entitled as “Genome-wide identification, exogenous hormone response, gene structure and conserved motif analysis of the GRF gene family in Cerasus humilis” for your consideration to be published in Biology.
Here is our response to the reviewer's comments. Thank you very much for reviewing our paper. We wish you and the reviewer a smooth work and look forward to hearing from you again.
Reviewer:
I have reviewed the manuscript, which presents a comprehensive study on the GRF gene family in response to exogenous hormones in Cerasus humilis. The methodology appears sound, and the results are well presented. Below is a list of reviewed comments.
Language and Grammar/Phrasing: The English is generally understandable, but there are numerous instances of grammatical errors, awkward phrasing, and typos that hinder readability. Professional English editing would significantly improve the manuscript.
Response: Thank you for your valuable advice.We have corrected grammar errors, wording mistakes, and typos in the manuscript to facilitate the reviewers' understanding of the manuscript.
Figures: Please add the detailed captions in the title of each figure for clear readability.
Response: Thank you very much indeed for your comments.We have added more detailed titles to each chart title for both reviewers and readers to read clearly.
Data-retrieving information: mention the exact date of retrieving the data from different databases.
Response: Thank you for your valuable advice.We have added the exact dates of the different databases mentioned in the manuscript for your review.
Write the complete names and versions of all software used for analysis.
Response: Thank you very much indeed for your comments.We have added the complete names and versions of all software used for analysis in the manuscript for your review.
Write the exact genome versions of all the crops used for evaluating the phylogenetic association.
Response: Thank you for your valuable advice.We have added the exact genome versions and search websites for all crops used to evaluate phylogenetic associations in the manuscript for your review.
Scientific names and gene names should be italicized.
Response: Thank you very much indeed for your comments.We have carefully reviewed and revised the manuscript to address the issue of scientific and genetic names being italicized for your review.
Abstract: Please mention the names and IDs of candidate genes and encoding proteins.
Response: Thank you for your valuable advice.We have indicated the names and IDs of candidate genes and encoded proteins in the manuscript abstract for your review.
In the Plant Material section, please add the complete details of exogenous GA hormone treatments.
Response: Thank you very much indeed for your comments.We have added complete and detailed information on exogenous GA hormone treatment in the Plant Materials section of the manuscript for your review, highlighted in blue font.
Discussion: It needs to be expanded by adding the more informative and comparative literature about genes' regulatory function in response to the GA treatments in other crops.
Response: Thank you for your valuable advice.We have added information and comparative literature on the regulatory functions of gene response to GA treatment in other crops in the manuscript discussion section for your review, highlighted in blue font.
Finally, The modifications we made in the manuscript are highlighted in blue font. Thank you very much.
Reviewer 2 Report
Comments and Suggestions for Authors
This article presents a comprehensive genome-wide identification and analysis of the GRF (Growth Regulation Factor) gene family in Cerasus humilis, an ecologically and economically valuable shrub native to China. One of its major strengths is the detailed and multi-layered approach taken, spanning gene identification, phylogenetic classification, cis-regulatory element analysis, and chromosomal localization. Although I have some comments:
The abstract is missing a take-home message.
The introduction is well-written and has sufficient information. However, the study's hypothesis is missing. I recommend adding the hypothesis of the study before the objective, and then explaining the objective.
The methodology is explanatory and up to the mark. But I would recommend adding some more detail in (Gene structure and motifs of ChGRF) heading.
The results are well written, and the figures are understandable.
The discussion looks fragmented. I would recommend using the hypothesis you will build in the introduction to clarify whether you have achieved the hypothesis or not.
Again conclusion is missing with take-home message for the readers.
Author Response
Dear Editorand Reviewers,
We respectfully submit our original research article entitled as “Genome-wide identification, exogenous hormone response, gene structure and conserved motif analysis of the GRF gene family in Cerasus humilis” for your consideration to be published in Biology.
Here is our response to the reviewer's comments. Thank you very much for reviewing our paper. We wish you and the reviewer a smooth work and look forward to hearing from you again.
Reviewer 2
This article presents a comprehensive genome-wide identification and analysis of the GRF (Growth Regulation Factor) gene family in Cerasus humilis, an ecologically and economically valuable shrub native to China. One of its major strengths is the detailed and multi-layered approach taken, spanning gene identification, phylogenetic classification, cis-regulatory element analysis, and chromosomal localization. Although I have some comments:
The abstract is missing a take-home message.
Response: Thank you very much indeed for your comments.We have summarized the important information in the abstract section of the manuscript and highlighted it in red font for your review.
The introduction is well-written and has sufficient information. However, the study's hypothesis is missing. I recommend adding the hypothesis of the study before the objective, and then explaining the objective.
Response: Thank you for your valuable advice.We have added research hypotheses in the introduction section of the manuscript and provided explanations in the text for your review. The modified parts are highlighted in red font.
The methodology is explanatory and up to the mark. But I would recommend adding some more detail in (Gene structure and motifs of ChGRF) heading.
Response: Thank you very much indeed for your comments.We have modified the title and added information such as Gene structure and motifs of ChGRF in the title. The modified part is highlighted in red font.
The results are well written, and the figures are understandable.
Response: Thank you for your valuable advice.I will carefully revise the problematic areas.
The discussion looks fragmented. I would recommend using the hypothesis you will build in the introduction to clarify whether you have achieved the hypothesis or not.
Response: Thank you very much indeed for your comments.We have carefully revised the discussion section of the manuscript and provided an explanation of the assumptions in the previous introduction for your review, highlighted in red font.
Again conclusion is missing with take-home message for the readers.
Response: Thank you for your valuable advice.We have resummarized and condensed the important information in the conclusion section of the manuscript to present our conclusion, which is clear to the reviewers and readers, and highlighted in red font for your review.
Reviewer 3 Report
Comments and Suggestions for Authors
The manuscript presents a genome-wide analysis of the GRF gene family in Cerasus humilis and its expression in response to exogenous gibberellin (GA). It is a well-structured and timely study. The findings contribute significantly to understanding GRF gene functions in this ecologically and nutritionally valuable plant. However, there are several minor language issues, formatting inconsistencies, and areas needing clarification.
Line 7:
“12 GRF genes were discovered...” → Consider revising for conciseness:
“Twelve GRF genes were identified in the C. humilis genome.”
-
Line 9:
“5 Subfamily” → should be “five subfamilies” -
Line 21:
“GRF gene are an important class of gene” → should be “GRF genes are an important class of genes” -
Line 37:
“Resembling the N-terminal end of SWI2/SNF2 in yeast...” → Consider simplifying or clarifying for a broader audience. -
Line 104:
“domainas well as the Pfam databas” → should be “domains as well as the Pfam database” -
Line 112-113:
“Clustal Xwas” → Typo. Should be “ClustalX was” -
Line 135:
“David database” → Better to specify “Database for Annotation, Visualization and Integrated Discovery (DAVID)” -
Line 182:
Typo in “subgroups (A - E) according to the evolution of the tree and species evolution.” → suggest rephrasing for clarity:
“...into five subgroups (A–E) based on phylogenetic and species evolutionary relationships.” -
Line 343:
“we conducted a comprehensive analysis of ChGRFs...” → Change to passive voice to maintain academic tone:
“A comprehensive analysis of ChGRFs was conducted...” -
Line 415:
Typo/missing space in “enhanced 415our comprehension” → should be “enhanced our comprehension” -
Line 76:
What was the duration of GA treatment and number of biological replicates used? Please clarify. -
Line 102–106:
Were the candidate GRFs validated manually after domain screening? More detail on the criteria for inclusion/exclusion would help. -
Line 130–134:
Were only the 2000 bp upstream regions considered for cis-element analysis? If yes, please clarify. -
Line 174:
How were isoelectric point (pI) and molecular weights calculated? Please mention the specific tool or formula. -
Line 345–346:
How did you determine expression levels in response to GA – was there normalization across tissues and time points?
The authors are encouraged to cite the following recent chapter in their manuscript:
Nawaz, T., Nelson, D., Saleem, A., Bibi, M., Joshi, N., Fahad, S., ... & Khan, I. (2025). Genetic Engineering for Crop Biofortification. In Crop Biofortification: Biotechnological Approaches for Achieving Nutritional Security Under Changing Climate (pp. 263–294).
This chapter provides valuable insights into the role of transcription factors, including GRFs, in enhancing crop traits through genetic engineering and hormone signaling pathways. Since the present study investigates GRF gene family members in Cerasus humilis and their responses to exogenous gibberellin, the citation is highly relevant. It offers a broader biotechnological context and supports the discussion on the role of GRFs in plant growth regulation, stress responses, and potential genetic manipulation strategies. Citing this work would not only strengthen the scientific foundation of the manuscript but also highlight the broader applications of GRF-related research in crop improvement under changing climatic conditions.
Overall, this study provides valuable insights into the GRF gene family in Cerasus humilis, combining genome-wide analysis with hormone response evaluation. With minor corrections related to language, formatting, and a few methodological clarifications, this manuscript is suitable for publication.
Comments on the Quality of English LanguageThe manuscript communicates the intended ideas well. However, with some minor grammatical corrections and improvements in sentence structure, the clarity and overall readability can be further enhanced.
Author Response
Dear Editorand Reviewers,
We respectfully submit our original research article entitled as “Genome-wide identification, exogenous hormone response, gene structure and conserved motif analysis of the GRF gene family in Cerasus humilis” for your consideration to be published in Biology.
Here is our response to the reviewer's comments. Thank you very much for reviewing our paper. We wish you and the reviewer a smooth work and look forward to hearing from you again.
Reviewer 3
The manuscript presents a genome-wide analysis of the GRF gene family in Cerasus humilis and its expression in response to exogenous gibberellin (GA). It is a well-structured and timely study. The findings contribute significantly to understanding GRF gene functions in this ecologically and nutritionally valuable plant. However, there are several minor language issues, formatting inconsistencies, and areas needing clarification.
Line 7:
“12 GRF genes were discovered...” → Consider revising for conciseness:
“Twelve GRF genes were identified in the C. humilis genome.”
Response: Thank you very much indeed for your comments.We have made modifications according to your feedback and highlighted them in green font.
Line 9:
“5 Subfamily” → should be “five subfamilies”
Response: Thank you for your valuable advice.We have made modifications according to your feedback and highlighted them in green font.
Line 21:
“GRF gene are an important class of gene” → should be “GRF genes are an important class of genes”
Response: Thank you very much indeed for your comments.We have made modifications according to your feedback and highlighted them in green font.
Line 37:
“Resembling the N-terminal end of SWI2/SNF2 in yeast...” → Consider simplifying or clarifying for a broader audience.
Response: Thank you for your valuable advice.We have consulted relevant literature on this sentence and made corrections for your review, which will be highlighted in green font in the text.
Line 104:
“domainas well as the Pfam databas” → should be “domains as well as the Pfam database”
Response: Thank you very much indeed for your comments.We have made modifications according to your feedback and highlighted them in green font.
Line 112-113:
“Clustal Xwas” → Typo. Should be “ClustalX was”
Response: Thank you for your valuable advice.We have made modifications according to your feedback and highlighted them in green font.
Line 135:
“David database” → Better to specify “Database for Annotation, Visualization and Integrated Discovery (DAVID)”
Response: Thank you very much indeed for your comments.Due to our negligence in using the wrong database David, we have revised this sentence and highlighted it in green font in the text for your review.
Line 182:
Typo in “subgroups (A - E) according to the evolution of the tree and species evolution.” → suggest rephrasing for clarity:
“...into five subgroups (A–E) based on phylogenetic and species evolutionary relationships.”
Response: Thank you for your valuable advice.We have made modifications according to your feedback and highlighted them in green font.
Line 343:
“we conducted a comprehensive analysis of ChGRFs...” → Change to passive voice to maintain academic tone:
“A comprehensive analysis of ChGRFs was conducted...”
Response: Thank you very much indeed for your comments.We have made modifications according to your feedback and highlighted them in green font.
Line 415:
Typo/missing space in “enhanced 415our comprehension” → should be “enhanced our comprehension”
Response: Thank you for your valuable advice.We have made modifications according to your feedback and highlighted them in green font.
Line 76:
What was the duration of GA treatment and number of biological replicates used? Please clarify.
Response: Thank you very much indeed for your comments.The spraying time of GA is 0 hours, 6 hours, 12 hours, 24 hours, 3 days, and 5 days, and biological replicates are performed 3 times to ensure reliable experimental data. Mark the modified information in green font.
Line 102–106:
Were the candidate GRFs validated manually after domain screening? More detail on the criteria for inclusion/exclusion would help.
Response: Thank you for your valuable advice.After domain name screening, we manually validated the candidate GRFs, removed duplicates, and retained the gene fragment with the longest gene sequence. The modified information is marked in green font.
Line 130–134:
Were only the 2000 bp upstream regions considered for cis-element analysis? If yes, please clarify.
Response: Thank you very much indeed for your comments.We only considered the upstream region of 2000 bp for cis component analysis.
Line 174:
How were isoelectric point (pI) and molecular weights calculated? Please mention the specific tool or formula.
Response: Thank you for your valuable advice.We use the ExPASy website to predict the physicochemical properties of proteins( http://web.expasy.org/protparam/ Visited on December 28, 2024, including the isoelectric point (PI) and molecular weight (MW) of ChGRFs.
Line 345–346:
How did you determine expression levels in response to GA – was there normalization across tissues and time points?
Response: Thank you very much indeed for your comments.Due to our negligence, we mistakenly wrote the expression level of GA in the text. We have revised the text to correctly indicate that the expression level of ChGRF gene under GA treatment varies at different time points. The revised information is highlighted in green font.
The authors are encouraged to cite the following recent chapter in their manuscript:
Nawaz, T., Nelson, D., Saleem, A., Bibi, M., Joshi, N., Fahad, S., ... & Khan, I. (2025). Genetic Engineering for Crop Biofortification. In Crop Biofortification: Biotechnological Approaches for Achieving Nutritional Security Under Changing Climate (pp. 263–294).
Response: Thank you for your valuable advice.We cited the literature appropriately in the text, which was very helpful to us.
Round 2
Reviewer 1 Report
Comments and Suggestions for Authors
In my opinion, the authors incorporated the suggested revisions in satisfactory form, and the revised manuscript is accepted. Thanks
Comments on the Quality of English LanguageThe English could be improved to more clearly express the research.